# Effects of Harvest Time on the Aroma of White Wines Made from Cold-Hardy Brianna and Frontenac Gris Grapes Using Headspace Solid-Phase Microextraction and Gas Chromatography-Mass Spectrometry-Olfactometry

**DOI:** 10.3390/foods8010029

**Published:** 2019-01-16

**Authors:** Somchai Rice, Madina Tursumbayeva, Matthew Clark, David Greenlee, Murlidhar Dharmadhikari, Anne Fennell, Jacek A. Koziel

**Affiliations:** 1Midwest Grape and Wine Industry Institute, Iowa State University, Ames, IA 50011, USA; somchai@iastate.edu (S.R.); murli@iastate.edu (M.D.); 2Interdepartmental Toxicology Graduate Program, Iowa State University, Ames, IA 50011, USA; 3Department of Agricultural and Biosystems Engineering, Iowa State University, Ames, IA 50011, USA; madina@iastate.edu; 4Department for Horticultural Science, University of Minnesota, St. Paul, MN 55108, USA; clark776@umn.edu; 5Tucker’s Walk Vineyard and Winery, Garretson, SD 57030, USA; dave@tuckerswalk.com; 6Department of Agronomy, Horticulture and Plant Science, BioSNTR, South Dakota State University, Brookings, SD 57006, USA; anne.fennell@sdstate.edu

**Keywords:** Frontenac gris, Brianna, wine aroma, SPME-GC-MS, olfactometry, cold-hardy grapes

## Abstract

The Midwest wine industry has shown a marked increase in growers, hectares planted, wineries, and wine production. This growth coincides with the release of cold-hardy cultivars such as Brianna and Frontenac gris, in 2001 and 2003, respectively. These white grape varieties account for one-third of the total area grown in the state of Iowa. It is generally accepted that the wine aroma profile plays a crucial role in developing a local, sustainable brand. However, the identity of Brianna/Frontenac Gris-based wine aromas and their link to the grape berry chemistry at harvest is unknown. This study aims to preliminarily characterize key odor-active compounds that can influence the aroma profile in wines made from Brianna and Frontenac gris grapes harvested at different stages of ripening. Brianna and Frontenac gris grapes were harvested approximately 7 days apart, starting at 15.4 °Brix (3.09 pH) and 19.5 °Brix (3.00 pH), respectively. Small batch fermentations were made for each time point with all juices adjusted to the same °Brix prior to fermentation. Odor-active compounds were extracted from wine headspace using solid-phase microextraction (SPME) and analyzed by gas chromatography-mass spectrometry (GC-MS) and simultaneous olfactometry (O). Over 30 odor-active compounds were detected. Aromas in Brianna wines developed from “cotton candy” and “floral”, to “banana” and “butterscotch”, then finally to “honey”, “caramel” and an unknown neutral aroma. Frontenac gris wines changed from an unknown neutral aroma to “fruity” and “rose”. Results from the lay audiences’ flavor and aroma descriptors also indicate a shift with harvest date and associated °Brix. To date, this is the first report of wine aromas from Brianna and Frontenac gris by GC-MS-O. Findings from this research support the hypothesis that aroma profiles of Brianna and Frontenac gris wines can be influenced by harvesting the grapes at different stages of ripening.

## 1. Introduction

The business of grapes and wine generated over $7.5 billion U.S. dollars (USD) in the upper Midwestern states of Iowa, Illinois, Michigan, Minnesota, North Dakota, South Dakota, and Wisconsin in 2017 [1]. This included the direct economic impact from vineyard and winery activities as well as tourism, resulting in over 110,000 jobs and over $3 billion in wages (Table 1).

In Iowa, the number of grape growers, vineyards of grapes, wineries, and wine production has increased in the last two decades (Figure 1 and Figure 2) [2]. In a report by Tuck and Gartner in 2014, 100 hectares of grapes planted in Iowa were of the cold-hardy white varieties [3]. These numbers were extrapolated from self-reported surveys to determine the baseline of activity involving cold-hardy grape varieties. Of this estimated 100 hectares, 27% of the plantings were Frontenac gris and Brianna varieties.

There is continuous interest in understanding the chemical origin of grape aromas. Our working hypothesis is that this information could help growers and winemakers to determine a more targeted harvest date, based on the desired aromas. It also would allow an assessment of how various winemaking practices influence aroma, an important factor of wine quality. This information could streamline the production of new grape varieties by permitting the selection of varieties showing certain aromatic attributes. Despite these advantages, determining the chemical origin of varietal aromatic character is complicated. First, odor-active compounds in grapes often occur in nonvolatile forms. These compounds are released only upon crushing [4], through yeast metabolism [5], or during aging [6]. A varietal character can originate from a combination of compounds and not from varietal specific compounds. Extraction procedures may influence the stability of odor-active compounds. Identification and quantification of odor-active compounds are needed to understand the aroma potential of new cold-hardy grape varieties.

The added benefit of simultaneous olfactometry (O) and chemical analysis (e.g., by gas chromatography-mass spectrometry (GC-MS)) allows for characterization of trace amounts of compounds with detection limits below that of the mass detector (i.e., 2-isobutyl-3-methoxypyrazine, a “green bell pepper” aroma with detection threshold less than 0 ppb) [7,8]. Brianna is known, at least anecdotally within the industry, to develop an unwanted “foxy” aroma if harvested after 14–16 °Brix. However, there is a lack of scientific data to support this observation.

Grape maturity levels expressed by sugar content (measured as °Brix) and titratable acidity (TA) in grape berries has a great impact on wine quality and aroma as well. During the period of grape ripening, sugar content increases in berries while the TA level decreases. The relationship between those two factors affects the release of wine odor-active compounds. As sugar increases and acidity decreases, the aroma of wine changes from “herbal” to “fruity” [9]. However, higher sugar content in berries resulting in higher ethanol production can decrease the volatility of odor-active compounds in wine, and fruity aromas change to alcohol-associated aromas [10]. Grapes are typically harvested when pH levels are between 3.2 to 3.4 for Brianna [11] and around 3.0 for Frontenac gris [12]. Brianna fruit has “grapefruit, tropical” and slight “floral” characteristics [11]. Frontenac gris fruit has aromas of “peach, apricot” and “tropical fruits” [13]. These cold-hardy cultivars were introduced to the public in 2001 (Brianna) [11] and 2003 (Frontenac gris) [12]. The cultivars are advantageous in cold climates, where *V. vinifera* will not survive the extreme low temperatures. Brianna was shown to be a top yielding cultivar among select cold-hardy cultivars with the lowest average titratable acidity [14].

There is a need to characterize aromas from these new cold-hardy cultivars in order to understand and improve the potential of the final product. The objective of this study was to preliminarily associate odor-active compounds in Brianna and Frontenac gris white wines with different stages of grape berry ripening (i.e., with increasing sugar content and pH). This was completed by analyzing odor-active compounds in the headspace of wine using solid-phase microextraction (SPME) and simultaneous chemical and sensory analysis using gas chromatography-mass spectrometry-olfactometry (GC-MS-O) [15,16].

In our previous research, we developed an automated headspace SPME-GC-MS-O method for aroma profiles of seven cold-hardy wines [15]. The effects of the SPME fiber type (7 coatings), the headspace SPME extraction time (10 distinct times from 10 s to 1 h), the extraction temperature (6 set points from 35 to 80 °C), the incubation time (5 set points for headspace equilibration from 0 to 20 min), the sample volume (4 set points from 1 to 4 mL in a 10 mL vial), the desorption time (6 set points from 30 to 300 s), and the salt addition (5 set points) were tested. We used the optimized SPME conditions from previous research [15] in this current work. A multivariate analysis was used to illustrate the effects of harvest time on wine odor-active compounds. There is a need to characterize aromas from these new cold-hardy cultivars in order to understand and improve the potential of the final product. This is the first report of odor-active compounds in wines made from Frontenac gris and Brianna grapes at different levels of maturity. Information from this study can guide growers and winemakers in optimizing winemaking techniques and harvest decisions. This (GC-O) technique has been used in wine aroma analysis in Chardonnay [17], Muscat [18], Cabernet Gernischt, Cabernet Sauvignon, Cabernet Franc, Merlot [19], and native American grapes (Vitis) [20].

## 2. Materials and Methods

### 2.1. Grape Samples Collection and Winemaking

The working hypothesis is that wine aromas are affected by Brianna and Frontenac gris berry maturation (i.e., change in pH and sugar content as °Brix) at the time of harvest. Brianna and Frontenac gris grapes were grown in a Tucker’s Walk vineyard in Garretson, South Dakota. Brianna and Frontenac gris grapes’ characteristics at harvest are given in Table 2. Tucker’s Walk produced the wines using the protocols developed for the Northern Grapes Project [21] during the 2015 growing season and are described as follows. Briefly, grapes were harvested approximately one week apart. Four small batches of Brianna and three small batches of Frontenac gris wines were made on-site, (*n* = 2), using the same winemaking process. Grapes (110 to 120 kg) were processed in a crusher/destemmer and pressed, and juice sugar content was adjusted to 24 °Brix for Frontenac gris and 20 °Brix for Brianna. Frontenac gris, a bud sport from Frontenac and a high acid grape, is typically harvested for commercial wine at 22–24 °Brix. Brianna, a low acid grape, is typically harvested between 16 and 20 Brix. Brianna and Frontenac gris juices were brought to 20 and 24 °Brix, respectively, at each harvest time point and fermented to dryness. This provided the same alcohol content in the respective cultivars across harvest dates. Inoculated juice was allowed to start fermenting at ambient temperature for 24 h, then immediately moved into 13 °C and fermented to dryness. The wines (14 total) were analyzed by chemical and sensory analysis in triplicate.

### 2.2. Informal Sensory Analysis of Brianna Wine by Wine Industry Professionals

Wines from each fermentation were analyzed in blind tastings by lay audiences at two viticulture and enology conferences (Minnesota Cold Climate Conference and Nebraska Vindemia). These panelists included grape growers, winemakers, vineyard/winery owners, and research scientists. Data was gathered from 32 and 23 individuals, respectively, and pooled for analysis. The panelists were asked to provide flavor descriptors and any wine quality notes. The descriptive terms were generated by the audience members and extracted from the data sheets. The terms were reduced from 78 to 61 terms by combining similar terms. For example, “citrus” includes lemon, lemongrass, grapefruit, and lime. The top 24 terms were selected as those having been mentioned by at least 4 panelists. A spider plot was created using the term’s incidence as the response variable.

### 2.3. Preparation of Wine Samples

A 10 mL glass amber vial with a magnetic screw top and polytetrafluoroethylene (PTFE)-lined septum was used. Undiluted wine samples and serial dilutions of wine samples in model wine (4 mL total volume) were prepared using dilution factors of 2, 4, 8, 16, and 32 [22]. The model wine was 5 mg/mL of potassium bitartrate in 12% ethanol in water. Two g of sodium chloride was added to each 10 mL vial to enhance headspace SPME extraction.

### 2.4. Automated SPME Extraction

A 50/30 µm divinylbenzene (DVB)/Carboxen/polydimethylsiloxane (PDMS) SPME fiber (Sigma-Aldrich, St. Louis, MO, USA) was used to extract and pre-concentrate odor-active compounds from the headspace of wine samples. A Leap Technologies CombiPal (Trajan Scientific, Pflugerville, TX, USA) was used for automated headspace sampling with the following parameters: 500 rpm agitation speed during incubation and extraction, 10 min incubation/extraction time at 50 °C, and 260 °C desorption for 2 min directly into the GC inlet. To prevent carryover between samples, the SPME fiber was also cleaned in a needle heater (260 °C for 2 min) under a flow of clean helium prior to each analysis.

### 2.5. Chemical and Sensory Analysis

An Agilent (7890B and 5977A) GC-MS was used for analysis, fitted with two columns in series. The first column was non-polar (BPX-2, 83 m × 530 µm × 0.5 µm, SGE-Trajan Scientific, Pflugerville, TX, USA) and pressure balanced at the midpoint with a second polar column (DB-WAXETR, 30 m × 530 µm × 0.25 µm, Agilent Technologies, Santa Clara, CA, USA). Effluent from the second column was split 1:3 by restrictor columns to the single quadrupole mass spectrometer and olfactometry sniff port, respectively (1 part to MS and 2 parts to O-port). The GC temperature profile was initially 40 °C, held for 3 min, 7 °C/min ramp to 220 °C, held for 11.29 min. Data acquisition was collected in full scan mode, the mass range was m/z 33 to 450, and the electron ionization energy was 70 eV. The instrument was tuned daily prior to analysis. MassHunter (v. B.07.00.1413, Agilent, Santa Clara, CA, USA) was used for mass spectral data acquisition and analysis. AromaTrax (v. 10.1, MOCON, Round Rock, TX, USA) was used for sensory data acquisition (i.e., the aromagram). Multitrax Multidimensional Control Software (v. 10.1, MOCON, Round Rock, TX, USA) was used for pressure balance programming. A single trained human panelist was used to assign aroma descriptors and intensity to each compound. This initial research on the popular two cold-hardy varieties was a “screening”-type work aiming to find odor-active compounds. At this (screening) stage, using one panelist is sufficient to achieve the stated aims, i.e., to preliminarily characterize odor-active compounds. This information should be used for follow-up studies as a starting point for proper experimental design. Since ethanol was expected to be present in each sample, the intensity of ethanol was assigned as 50 on a 1 to 100 intensity scale. This process has been described in detail elsewhere [22,23].

### 2.6. Data Analysis

Odor-active compounds collected from wine headspace was tentatively identified by matching mass spectra to the NIST11 library, Wiley 6N library. All compounds with 80% spectral match or higher and above the 1000 peak area count threshold were considered for the analyses. Aroma descriptors from the panelist were compared to known aroma descriptors for additional verification. The matching of retention time indices was not appropriate in this case due to the GC columns of different polarity in-series configurations. The identification of compounds by the analysis of the pure standard was not performed, but the specific ions of a compound are provided in Table A1, when present in the chromatogram above the threshold.

Aroma extract dilution analysis (AEDA) was used to identify the most important compounds. From the aromagram, the odor dilution (OD) of each aroma event was multiplied by the measured intensity value resulting in the weighted intensity. This data was plotted with intensity (% full-scale) vs. time. Compounds with a higher OD were considered to be major contributors to the aroma profile of the wine.

Aroma descriptor intensity and OD were analyzed by principal components analysis (PCA) and cluster summary analysis using JMP Pro 12.0.1 (SAS, Cary, NC, USA). PCA is useful in summarizing all the odor-active compounds, detected by the human nose, in the wines among all conceivable linear combinations. A cluster summary analysis was also performed to determine the most representative aroma compound (i.e., the cluster variable with the largest squared correlation with its cluster component).

## 3. Results

Aroma events were simultaneously recorded using the sniff port by a trained human panelist during chromatographic analysis. A summary of the aroma events and the tentative identification by mass spectra is given in Table A1 in Appendix A. There were 57 unique aroma events detected by olfactometry and 32 odor-active compounds tentatively identified by mass spectrometry in Frontenac gris and Brianna wines. There were 35 and 34 aroma events recorded for Frontenac gris and Brianna wines, respectively. Aroma descriptors that were common between Frontenac gris and Brianna wines included “alcoholic, banana, body odor, butterscotch, cut grass, floral, fruity, garlic, honey, caramel, overripe fruit, rose, rotten eggs, solvent, strawberry”, and “tomato”. Aroma descriptors unique to Frontenac gris wines included “woody, carrots, cereal, mushroom, sweaty”, and “vinegar”. Aroma descriptors unique to Brianna wines included “barnyard, cotton candy, and mint.” The intensity of aromas (detailed in Materials and Methods section) in Frontenac gris and Brianna wines, according to harvesting parameters, is summarized in Table 3.

Seventeen aromas did not yield suitable (>80%) corresponding mass spectral matches and are labeled as “unknown”. This could indicate that the compound responsible for this aroma is not concentrated enough for the mass detector to respond and that the odor detection threshold for this compound was very low. The evidence that the human nose can be more sensitive than the chemical detector is consistent with the notion that simultaneous chemical and sensory analyses are useful for analyses of complex wine headspace. Wine headspace aroma is one of the first attributes experienced by consumers and wine enthusiasts.

### 3.1. Frontenac Gris White Wine Aroma Analysis by SPME-GC-MS-O

White wines from Frontenac gris grapes had 35 recorded aroma events across all samples. Aromas of “honey, caramel, butterscotch” and “strawberry, honey” had no variation in odor dilution (OD, defined in Methods) and were not used in the multivariate analysis. The aromas with the highest intensity in the Frontenac gris wines were “banana”, “fruity 2”, “honey”, and “unknown neutral 1”. Cluster summary analysis of OD showed that “rotten eggs, sulfury” and “unknown neutral 1” were the most representative aromas in these Frontenac gris wine. A “rotten eggs” smell in wine is considered a wine fault due to the winemaking process and therefore not considered a characteristic aroma of the grape. A chromatographic peak was not present at the corresponding retention time for “unknown neutral 1”. As pH and sugar accumulation in the berry increased, key odor-active compounds in these Frontenac gris wines developed from “unknown neutral 2” and “fruity 1” to “rose 1” (Figure 3). These correspond to mass spectral matches of “unknown neutral 2” to decanoic acid (CAS 334-48-5) and “fruity 1” to ethyl methylbutyrate (CAS 7452-79-1). A suitable mass spectral match was not found for the identification of “rose 1.” An open source aroma database [7] lists the percepts of “rancid, fat” for decanoic acid and “apple, characteristic of Golden delicious” for ethyl methylbutyrate. In the Flavornet database, 16 different compounds are listed with the aroma descriptor “rose”.

### 3.2. Brianna White Wine Aroma Analysis by SPME-GC-MS-O

White wine from Brianna grapes had 34 recorded aroma events across all samples. The “rotten eggs” aroma had no variation in OD and was not used in the multivariate analysis similarly to Frontenac gris. The most intense aromas in these Brianna wines were “overripe fruit 2”, “floral”, and “unknown neutral 5”. The most representative aromas, as indicated by cluster analysis, in these Brianna wines were “banana”, “floral”, “honey, caramel”, “butterscotch 1”, “tomato 1”, and “overripe fruit 2”. Corresponding compounds from mass spectral searches are isoamyl acetate (banana, CAS 123-92-2), ethyl isobutyrate (“honey, caramel”, CAS 97-62-1), and isoamyl alcohol (“overripe fruit 2”, CAS 123-51-3). A suitable mass spectral match was not found for the “floral” aroma compound. A chromatographic peak was not recorded corresponding to “butterscotch 1”, although the database lists methyl vanillate [7] as a source of this aroma. Two mass spectral matches were identified for “tomato 1”: diphenylmethane (“green”, CAS 101-81-5) [24] and isobutyl decanoate (“fermented”, CAS 30673-38-2) [24]. The “floral” aroma is associated with 48 different compounds [7]. As pH and sugar accumulation in these Brianna berries increased, key odor-active compounds for each harvest changed (Figure 4). When harvested at the lowest sugar content and pH, the wines had a “cotton candy” (ethyl decanoate, CAS 110-38-3) and “floral” aroma. From 17.6 to 18.6 °Brix, aromas changed from “banana” to “butterscotch.” At the highest sugar and pH, the key aromas in the Brianna wines were “honey, caramel” and “unknown neutral 1” (isobutyl alcohol, CAS 78-83-1). This change in aromas over Brianna berry ripening is shown in Figure 4.

## 4. Discussion

SPME has been used to quantify volatile by-products in industrial ethanol [25], volatile cogeners in food-grade ethanol [23], and volatile odor-active compounds in cold-hardy wines made from Marquette and Frontenac [22] and even used to characterize street drug aromas [26,27,28]. Odor-active compounds in wine headspace must be extracted quickly and efficiently in order to minimize the effects of oxidation on the wine aroma profile. In this research, a SPME 50/30 µm divinylbenzene (DVB)/Carboxen/polydimethylsiloxane (PDMS) coating was suitable for extraction of a wide variety of aroma volatiles including alcohols, esters, aldehydes and ketones, phenolics, and acids. Ethanol being the most prevalent in headspace did not outcompete volatile aromas for SPME sorption sites.

Simultaneous sensory and chemical analyses of white wine aroma was facilitated by the use of GC-MS-O. The advantage of using olfactometry (O) simultaneously with chemical detection is the ability to focus on selected fewer aroma-causing compounds present in a very complex mixture of the wine headspace matrix. A sole focus on chemical analyses can preclude finding the aroma-defining volatile compounds in wine.

Grape sugar content (°Brix) varies depending on the species, variety, maturity (ripening), and health of the fruit [10]. Cultivars of European *Vitis vinifera* generally accumulate sugar at a concentration of 20% or more at maturity [29]. The cold-hardy cultivars Brianna and Frontenac gris pedigree includes *V. riparia*, *V. labrusca*, and *V. vinifera* [30,31]. Brianna, in particular, is often harvested at a lower °Brix to avoid “foxy” flavors. Sugar is added (chaptalization) to the juice to develop the 10–12% alcohol content typical of most still (non-sparkling) table wines [32]. The effects of sugar content and ethanol concentrations on the sensory attributes of young and aged sweet wines is found elsewhere [33,34]. However, there are few intervention options for enhancing the desired aromas. Thus, wine cold-hardy white wines produced from European/native N. American cultivars such as Brianna and Frontenac gris need to be “farmed for flavor.” This means that growers should consider an optimal flavor profile as a harvest parameter, in addition to the °Brix, pH, and TA.

Grapes produce few aldehydes significant in varietal aromas. This may result from their reduction to alcohols during primary fermentation. Of the aldehydes not metabolized during primary fermentation, C-6 aldehydes appear to be the most noteworthy [35]. These aldehydes are responsible for the grassy to herbaceous odor associated with certain grape varieties or with wines made from immature grapes. They appear to be formed during crushing by the enzymatic oxidation of grape lipids [4]. Most aldehydes found in wine are created during processing or fermentation or are extracted from oak cooperage [32].

Likewise, few ketones are found in grapes. The norisoprenoid ketones (i.e., beta-damascenone, alpha-ionone, and beta-ionone) are persistent throughout fermentation [32]. The “apple, rose, honey” aroma of beta-damascenone [7] and low odor threshold [24] imply that it is important in the aroma of several grape varieties including “Chardonnay” [36] and “Riesling” [37]. The “seaweed, violet, flower, raspberry” aroma of beta-ionone [7], along with beta-damascenone, are important in the aroma of several red grape varieties [38]. Other ketones that are generated by fungal metabolism or produced during fermentation and acetals produced during aging and distillation will not be discussed in this research.

Of all the aromatic constituents of wine, esters are the most abundant. Most of these esters are found only in trace amounts and have either low volatility or non-distinct odors, and their importance to wine fragrance is often discounted. The more common esters such as acetate esters are derived from acetic acid and fusel alcohols, and the ethyl esters are formed between ethanol and fatty acids or nonvolatile, fixed organic acids. The fruity aromas are important in the aroma profile of young white wines [39]; however, the esters to the aromas of red wines is less understood.

Terpenes are an important group of aromatic compounds characterizing the aromas of “flower and lavender” (linalool), “rose and geranium” (geraniol), “sweet” (nerol), “oil, anise, mint” (alpha-terpineol), and “hyacinth” (hotrienol) [7]. Terpenes are responsible for the fragrance of herb-flavored wines such as vermouth and fruit-flavored wines. In addition, terpenes also characterize some wine grape cultivars, most notably the “Muscat” and “Riesling” families [40].

Pyrazines are important to the characteristic varietal aromas of several cultivars [41]. Ethyl 3-mercaptopropionate is an important compound suspected to be the “foxy” odor of some *V*. *labrusca* varieties [42]. Most thiols generate off-odors, and only a few contribute to the characteristic varietal aroma of wine grape cultivars. These are 4-mercapto-4-methylpentan-2-ol (“floral, lemon grapefruit”) [24] and 3-mercaptohexan-1-ol (“grapefruit”) [43]. Both compounds are important in the varietal character of “Sauvignon Blanc” [43]. A key aroma important in “Scheurebe” is 4-mercapto-4-methylpentan-2-ol [44].

Despite the information available on volatile wine odor-active compounds and their sensory perceptions, experienced tasters are not always able to determine the grape variety (Vinifera), even when 100% of the wine is made from that cultivar [45]. A review of wine aroma in grapes is provided elsewhere [46]. The question remains if these new cold-hardy cultivars produce a distinct varietal aroma in white wines. This research adds a valuable initial report on white wine aromas from Brianna and Frontenac gris grapes. To date, the only other published research on cold-hardy wine aromas pertains to red wines [47,48,49,50] and white wines [51]. Therefore, this research serves as a starting point for determining the odor-active compounds in Brianna and Frontenac gris cold-hardy wines. At this (screening) stage, using one panelist achieved the stated aims, i.e., preliminarily characterized odor active compounds. This information should be used for follow-up studies as a starting point for proper experimental design. This relatively low number of publications on cold-hardy wine varieties is significant compared with active research in Vinifera [52,53,54,55,56,57,58,59,60,61,62].

Results (obtained with GC-MS-O approach) from this research could be used to inform cold-hardy grape growers on “farming for flavor.” A shift of the aroma profile from “fruity 1” to “rotten eggs, sulfury” to “rose 1” was observed in wines made from Frontenac gris harvested at 19.5, 23.1, and 23.6 °Brix, respectively (Figure 3). The must was not submitted to cold-settling and might be a major reason for the “sulfury, rotten egg” odors found in the research wines. In addition, a shift of the aroma profile from “cotton candy” to “banana” to “floral” to “butterscotch” was observed in wines that were made from Brianna grapes harvested at 15.4, 17.6, 18.6, and 19.6 °Brix, respectively (Figure 4).

Similar shifts of actual flavor and aroma of wines made from Brianna were also observed during tasting sessions at conferences for wine industry professionals. Results from the lay audiences’ flavor and aroma descriptors (Figure 5) also indicate a shift with harvest date and associated °Brix. The most obvious change at the late harvest date is the use of the term “foxy”, a negative characteristic associated with *V. labrusca*-based wines. There was also a decrease in the use of “acidity,” although “citrus” was still mentioned. Additional flavor descriptors that had a higher incidence in the late harvested wine included “bitter”, “floral”, and “pineapple”. The lay audiences’ perceptions of the Brianna wine detected some of the “sulfur”, “dirty”, “musty” aromas but at a very low incidence.

This research will help support the sustainable development of cold-hardy grape growing and the winemaking industry in Midwest U.S by providing a baseline for viticultural and wine-making practices. The next logical step would be to relate aroma-active compounds with sensory attributes by means of pattern recognition techniques that use multivariate statistical tests such, as principal component analysis, cluster analysis, or even partial least square (PLS) algorithms as previously described [63,64]. It is also possible to use the volatile data obtained by GC to construct odorant series with a given odor activity value for comparison purposes with sensorial data as in References [26,27,28,65,66,67].

More research is warranted on the aromas of white wines produced from cold-hardy cultivars. Several recommendations could be made including repeated studies involving a greater number of growing seasons and eventually developing consistent regional wine styles. This could include linking the sensory characteristics such as color, body and mouthfeel [68], and aroma.

## 5. Conclusions

This is the first report of white wine aromas from cold-hardy Brianna and Frontenac gris by GC-MS-O. Findings from this research support the hypothesis that aroma profiles of Brianna and Frontenac gris wines can be influenced by harvesting the grapes at different stages of ripening. Evaluation of the respective cultivar wines from different harvest dates but the same alcohol content allowed the detection of over 30 odor-active compounds in the wine headspace for both Brianna and Frontenac gris. The particular wine aroma profile changed depending on the time of harvest and grape maturity. Aromas in Brianna wines developed from “cotton candy” and “floral” to “banana” and “butterscotch” and then finally to “honey”, “caramel”, and an “unknown neutral” aroma. Over 68% of the variation in harvest time was correlated with key odor-active compounds. Aromas in Frontenac gris wines changed from an “unknown neutral” aroma to “fruity” to “rose”. Over 98% of the variation in harvest time was correlated with key odor-active compounds. Wine tasting data generated by wine industry professionals at conferences showed a shift in flavor and aroma descriptors for Brianna wines. The shift of flavor and aroma descriptors is associated with the increase in °Brix and “foxy,” a negative characteristic associated with *V. labrusca*-based wines at the latest harvest dates. This research provides both positive and negative aroma characteristics associated with increased ripeness and will help support the sustainable development of cold-hardy grape growing and the winemaking industry in Midwest U.S by providing a baseline for viticultural and wine-making practices.

## Figures and Tables

**Figure 1 foods-08-00029-f001:**
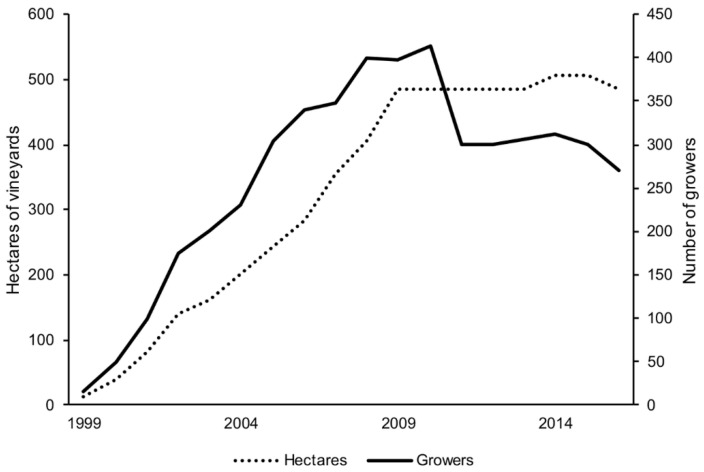
The increase in hectares of wine grapes and the number of growers in Iowa [2].

**Figure 2 foods-08-00029-f002:**
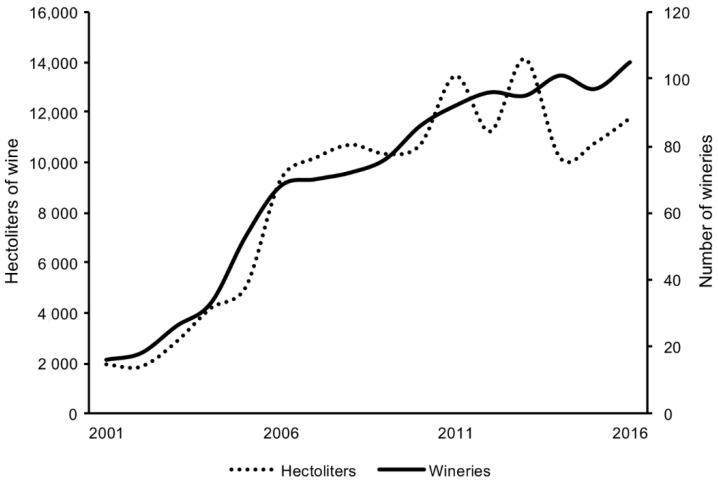
The increase in wine production (hectoliters) and a number of wineries in Iowa [2].

**Figure 3 foods-08-00029-f003:**
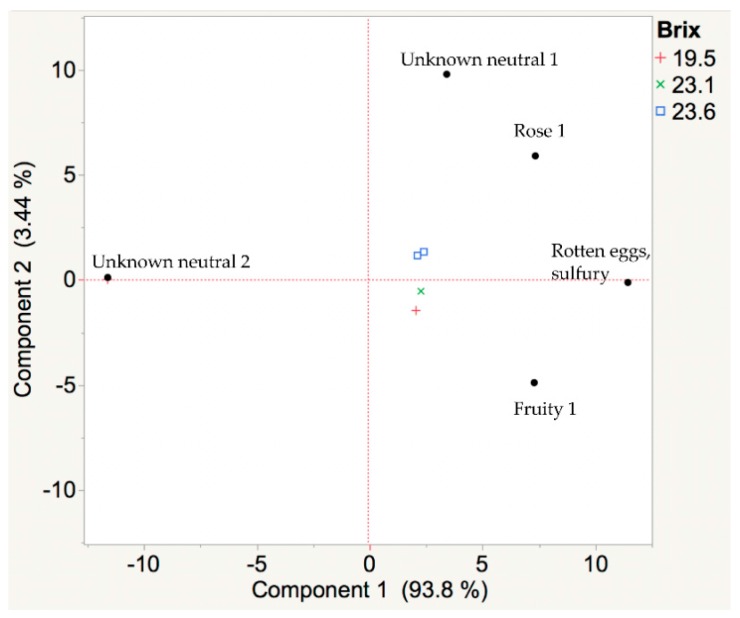
A principal components analysis (PCA) biplot of volatiles from the aroma extract dilution analysis of Frontenac gris wines made from berries harvested at three different ripening stages. Wines were made from Frontenac gris cold-hardy grapes harvested at 19.5, 23.1, and 23.6 °Brix. The juice was adjusted to 24 °Brix prior to fermentation. Wine headspace samples were collected by solid-phase microextraction (SPME) and analyzed with gas chromatography-mass spectrometry-olfactometry (GC-MS-O). Aroma descriptors were recorded by a trained human panelist. A shift of the aroma profile from “fruity 1”to “rotten eggs, sulfury” to “rose 1” was observed. Over 98% of the variation in harvest time is correlated with key odor-active compounds.

**Figure 4 foods-08-00029-f004:**
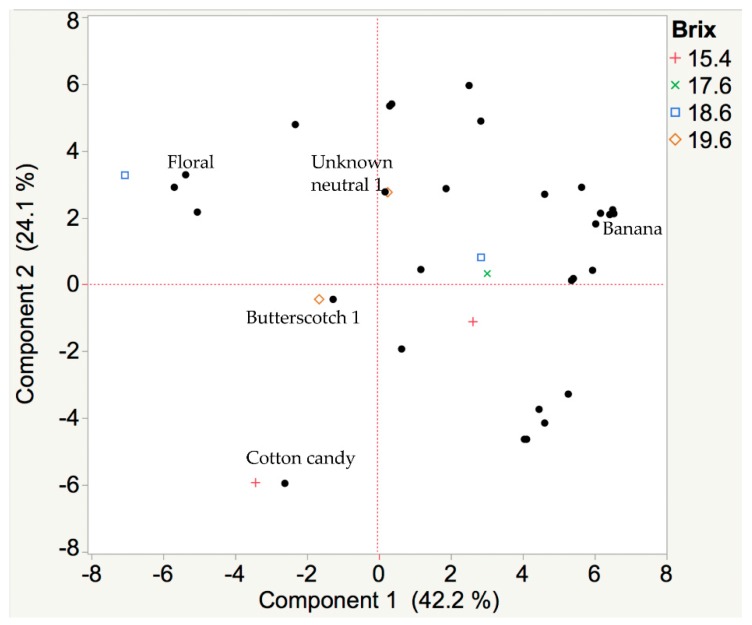
A PCA biplot of aromas from Brianna wines made from berries harvested at four different ripening stages. Wines were made from Brianna cold-hardy grapes harvested at 15.4, 17.6, 18.6, and 19.6 °Brix. The juice was adjusted to 20 °Brix for all time points prior to fermentation. Wine headspace samples were collected by SPME and analyzed with GC-MS-O. Aroma descriptors were recorded by a trained human panelist. A shift of the aroma profile from “cotton candy” to “banana” to “floral” to “butterscotch” was observed. Over 68% of the variation in harvest time is correlated with key odor-active compounds.

**Figure 5 foods-08-00029-f005:**
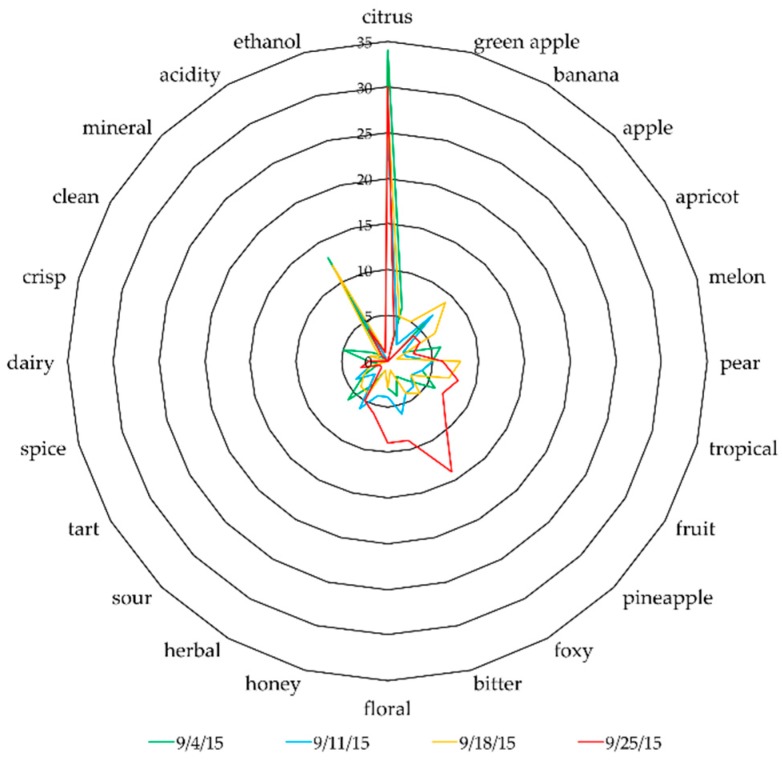
A spiderplot indicating a shift of flavor and aroma descriptors of Brianna wines, that were made from grapes harvested from 4 September to 25 September, from the wine tasting panels generated by lay audiences at conferences for wine industry professionals. A shift of flavor and aroma descriptors is associated with the increase in °Brix and appearance of “foxy”, a negative characteristic associated with *V. labrusca*-based wines at the latest harvest date.

**Table 1 foods-08-00029-t001:** Economic impact of the U.S. Midwest (cold climate) wine industry [1].

State	Economic Impact ^1^	Jobs	Wages ^1^	Vineyard Activity ^1^	Winery Activity ^1^	Tourism ^1^	Other ^1,2^
North Dakota	$135	2340	$57.3	$0.00680	$7.09	$0.245	$127
South Dakota	$180	2690	$62.4	$0.0719	$25.7	$1.69	$153
Iowa	$573	8760	$197	$1.10	$110	$29.0	$433
Minnesota	$979	15,400	$408	$1.22	$83.5	$21.3	$873
Wisconsin	$1320	20,700	$519	$1.12	$146	$39.7	$1130
Michigan	$1890	25,800	$710	$7.77	$325	$149	$1410
Illinois	$2480	34,800	$1060	$1.82	$247	$222	$2010
Totals	$7550	11,0000	$3010	$13.1	$944	$463	$6130

^1^ Millions of U.S. Dollars (USD); ^2^ includes wholesale, retail, associations, research, and education.

**Table 2 foods-08-00029-t002:** Brianna and Frontenac gris grapes’ characteristics at harvest.

Cultivar	Harvest Date	Berry °Brix	Berry pH
Frontenac gris	24 September 2015	19.5	3.00
Frontenac gris	1 October 2015	23.1	3.06
Frontenac gris	9 October 2015	23.6	3.18
Brianna	4 September 2015	15.4	3.09
Brianna	11 September 2015	17.6	3.19
Brianna	18 September 2015	18.6	3.29
Brianna	25 September 2015	19.6	3.45

**Table 3 foods-08-00029-t003:** Summary of the ranked weighted intensity of aromas (recorded by olfactometry) in wine made from Frontenac gris and Brianna grapes harvested at different stages of ripening. All juice was brought to 24 °Brix for Frontenac gris and 20 °Brix for Brianna prior to fermentation to ensure similar alcohol content in the wines from the different time points.

Cultivar	Berry °Brix	Berry pH	Aroma Descriptors (Weighted Intensity ^1^)
Frontenac gris	19.5	3.00	unknown pleasant (19), floral/fruity (11), floral (5), overripe (3), butterscotch (2), tomato (1), unknown pleasant 1 (0), unknown neutral 2 (0)
Frontenac gris	23.1	3.06	honey/caramel/butterscotch (431), fruity (419), cut grass/fruity (417), alcoholic (391), banana (382), body odor (359), fruity 1 (345), solvent (337), unknown pleasant (324), rose 2 (321), garlic (207), carrots/woody (178), cereal (152), honey (122), vinegar (57), woody 1 (55)
Frontenac gris	23.6	3.18	strawberry (524), strawberry/honey (395), sweaty (384), fruity 2 (244), match/sulfury (183), rose 1 (132), fecal (117), woody 2 (102), rotten eggs/sulfury (78), mushroom (63), unknown neutral 1 (5)
Brianna	15.4	3.09	rose (158), body odor (123), barnyard (122), butterscotch 2 (115), unknown pleasant (111), unknown neutral 2 (98), matchstick (92), mint (67), cotton candy (13)
Brianna	17.6	3.19	alcoholic (420), overripe fruit 2 (373), rotten eggs (106)
Brianna	18.6	3.29	strawberry 2 (579), fruity 3 (506), cut grass (500), floral/fruity (472), honey/caramel (468), banana (467), overripe fruit 1 (455), solvent (425), strawberry 1 (382), unknown neutral 3 (363), fruity 2 (316), garlic (239), unknown pleasant (193), fruity 1 (21), floral (5)
Brianna	19.6	3.45	tomato 2 (196), unknown neutral 4 (194), unknown neutral 5 (180), tomato 1 (179), unknown neutral 1 (59), fruity 4 (45), butterscotch 1 (3)

^1^ Defined in the Materials and Methods (Section 2.6).

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
