# Peer review of "Effects of Harvest Time on the Aroma of White Wines Made from Cold-Hardy Brianna and Frontenac Gris Grapes Using Headspace Solid-Phase Microextraction and Gas Chromatography-Mass Spectrometry-Olfactometry"

_foods, 2019, doi:10.3390/foods8010029_

Round 1

Reviewer 1 Report

The science presented is of a high quality. This is a substantial manuscript that occupies the bounds between biochemistry and food chemistry. The article is generally well written and will serve as a valuable resource for the characterization of aroma-active compounds. I suggest going ahead in a future in such a research line focusing on other minor components affecting food quality.

Anyway, I feel the authors should relate aroma-active components with sensory attributes by means of pattern recognition techniques that use multivariate statistical tests such as principal component analysis, cluster analysis, etc., or even partial least square (PLS) algorithms as in the following papers:

-“Relationships between Godello white wine sensory properties and its aromatic fingerprinting obtained by GC-MS. Food Chemistry, 2011, 129, 890-898”.

-“Aroma profile of Garnacha Tintorera-based sweet wines by chromatographic and sensorial analyses. Food Chemistry, 2012, 134(4), 2313-2325”.

It is also possible to use the volatile data obtained by gas chromatography to construct odorant series with a given odor activity value for comparison purposes with sensorial data as in:

-“Aroma potential of Brancellao grapes from different cluster positions. Food Chemistry, 2012, 132(1), 112-124”.

-“Active odorants in Mouraton grapes from shoulders and tips into the bunch. Food Chemistry, 2012, 133(4), 1362-1372”.

-“Floral, spicy and herbaceous active odorants in Gran Negro berries from shoulders and tips into the cluster, and comparison with Brancellao and Mouratón varieties. Food Chemistry, 2012, 135(4), 2771-2782.

Some other related papers or reviews are:

-. “Effects of sugar concentration processes in grapes and wine aging on aroma compounds of sweet wines – a review. Critical Reviews in Food Science and Nutrition, 2015, 55, 1053-1073”.

-. “Sensory Quality Control of Young vs. Aged Sweet Wines Obtained by the Techniques of Both Postharvest Natural Grape Dehydration and Fortification with Spirits During Vinification. Food Analytical Methods, 2013, 6(1), 289-300”.

-. “Evolution of the aromatic profile in Garnacha Tintorera grapes during raisining and comparison with that of the naturally sweet wine obtained. Food Chemistry, 2013, 139(1-4), 1052-1061”.

-. “Wine aroma compounds in grapes: a critical review. Critical Reviews in Food Science and Nutrition, 2015, 55(2), 202-218.”

The subject of the paper is very interesting.

Author Response

Comments and Suggestions for Authors

The science presented is of a high quality. This is a substantial manuscript that occupies the bounds between biochemistry and food chemistry. The article is generally well written and will serve as a valuable resource for the characterization of aroma-active compounds. I suggest going ahead in a future in such a research line focusing on other minor components affecting food quality.

Anyway, I feel the authors should relate aroma-active components with sensory attributes by means of pattern recognition techniques that use multivariate statistical tests such as principal component analysis, cluster analysis, etc., or even partial least square (PLS) algorithms as in the following papers:

-“Relationships between Godello white wine sensory properties and its aromatic fingerprinting obtained by GC-MS. Food Chemistry, 2011, 129, 890-898”.

-“Aroma profile of Garnacha Tintorera-based sweet wines by chromatographic and sensorial analyses. Food Chemistry, 2012, 134(4), 2313-2325”.

Authors’ response: We added these references and information in Discussion (immediately under Figure 5) as a commentary on next logical steps involving multivariate statistical tests.

It is also possible to use the volatile data obtained by gas chromatography to construct odorant series with a given odor activity value for comparison purposes with sensorial data as in:

-“Aroma potential of Brancellao grapes from different cluster positions. Food Chemistry, 2012, 132(1), 112-124”.

-“Active odorants in Mouraton grapes from shoulders and tips into the bunch. Food Chemistry, 2012, 133(4), 1362-1372”.

-“Floral, spicy and herbaceous active odorants in Gran Negro berries from shoulders and tips into the cluster, and comparison with Brancellao and Mouratón varieties. Food Chemistry, 2012, 135(4), 2771-2782.

Authors’ response: We added these references and information in Discussion (immediately under Figure 5) as a commentary on next logical steps involving multivariate statistical tests.

Some other related papers or reviews are:

-. “Effects of sugar concentration processes in grapes and wine aging on aroma compounds of sweet wines – a review. Critical Reviews in Food Science and Nutrition, 2015, 55, 1053-1073”.

-. “Sensory Quality Control of Young vs. Aged Sweet Wines Obtained by the Techniques of Both Postharvest Natural Grape Dehydration and Fortification with Spirits During Vinification. Food Analytical Methods, 2013, 6(1), 289-300”.

-. “Evolution of the aromatic profile in Garnacha Tintorera grapes during raisining and comparison with that of the naturally sweet wine obtained. Food Chemistry, 2013, 139(1-4), 1052-1061”.

Authors’ response: We added these references and information in Discussion as a commentary on sugar content and ethanol concentration.

-. “Wine aroma compounds in grapes: a critical review. Critical Reviews in Food Science and Nutrition, 2015, 55(2), 202-218.”

Authors’ response: We added these references and information in Discussion as a commentary on wine aroma in Vinifera.

The subject of the paper is very interesting.

Authors’ response: thank you for this comment.

Reviewer 2 Report

GENERAL REMARKS

INTRODUCTION

Give some information about Brianna and Frontenac gris grape varieties

Give some information about the techniques used e.g SPME for analysis of varietal aroma, GC-O for aroma analysis

CONCLUSIONS

the authors are advised to write conclusions in form of a text, no bullets

LINE 47: in the phrase "acres of grapes" change "grapes" to "vineyards"

line 59: "continuous" instead of "continued" would be more appropriate

line 63: what do the authors mean by "lines" in the phrase "by permitting the selection of lines" ?

line 64: change "varietal character" to "varietal aromatic character"

line 75: define better "later". Later than what? the usual harvest date? the optimum harvest date?

line 77: start with "Grape maturity level expressed by sugar content.... has (instead of have).."

lines 79-80" "The balance between those two factors affects the release of wine odor active compounds."  It is not exactly the balance, it is the maturation process.

lines 84, 85. The authors mean Brianna and Frontenac gris grapes, wines, or both?

lines 111-114 : " Multivariate analysis was used to illustrate the effects of harvest time on wine odor active compounds. This is the first report of wine aromas made from Frontenac gris and Brianna grapes at different levels of fruit maturity. Information from this study can guide growers and winemakers in optimizing winemaking techniques and harvest decisions"  HAS ALREADY BEEN WRITTEN IN PREVIOUS PARAGRAPH

lines 117-118: "berry maturation chemistry". Maybe the authors mean "berry maturation process"

lines 120-121: "Northern Grapes Project". Give a reference for that

line 124: how many kgs? of grapes

line 124: correct "distemmer" to "destemmer"

lines 124-125: why was the sugar content adjusted to different brix levels?

lines 124-125: If it is well understood, the must wasn't submitted to cold-settling after pressing and prior to fermentation. This might be a major reason for the "sulfuric, rotten egg" odors found in wines

line 128: Table Title. Correct to "Brianna and Frontenac Gris grapes characteristics at harvest"

Also correct in the text when referenced.

Line 149: was agitation performed during incubation or prior to? explain better

line 302: "throughout fermentation are" omit "are"

line 328: correct "are not able to" to "are not ALWAYS able to'"

LINE 331: "To date, the only other published research on cold-hardy wine aromas pertains to red wines [41-44]."

the authors need to take into consideration a master thesis

 "Descriptive analysis of Frontenac gris and Brianna wine grape and wine varieties "
A Thesis SUBMITTED TO THE FACULTY OF UNIVERSITY OF MINNESOTA
BY
Jenna M. Brady
December 2017

https://conservancy.umn.edu/bitstream/handle/11299/194662/Brady_umn_0130M_18804.pdf?sequence=1

Author Response

Comments and Suggestions for Authors

GENERAL REMARKS

INTRODUCTION

Give some information about Brianna and Frontenac gris grape varieties

Authors’ response: We added reference to work by Atucha et al. (2018).

Give some information about the techniques used e.g SPME for analysis of varietal aroma, GC-O for aroma analysis

Authors’ response: We added 11 new references (some of which address this issue) as a response to Reviewer 1. For example (Pawliszyn, 2009) has a comprehensive summary of SPME used in food and beverage industry. Several added references refer to GC-O research on wine aromas.  Also, we already had some background material on SPME and GC-O research in Discussion.

CONCLUSIONS

the authors are advised to write conclusions in form of a text, no bullets

Authors’ response: We made editorial changes to Conclusions to make them clearer and turn them into text.  

LINE 47: in the phrase "acres of grapes" change "grapes" to "vineyards"

Authors’ response: Changed “grapes” to “vineyards”

line 59: "continuous" instead of "continued" would be more appropriate

Authors’ response: Changed “continued” to “continuous”

line 63: what do the authors mean by "lines" in the phrase "by permitting the selection of lines" ?

Authors’ response: Changed “lines” to “varieties”.

line 64: change "varietal character" to "varietal aromatic character"

Authors’ response: Changed “varietal character” to “varietal aromatic character”

line 75: define better "later". Later than what? the usual harvest date? the optimum harvest date?

Authors’ response: Changed “later” to “after 14-16 °Brix”

line 77: start with "Grape maturity level expressed by sugar content.... has (instead of have).."

Authors’ response: Changed line 77 to “Grape maturity level expressed by sugar content (measured as °Brix) and titratable acidity (TA) in grape berries has a great impact on wine quality and aroma as well.”

lines 79-80" "The balance between those two factors affects the release of wine odor active compounds."  It is not exactly the balance, it is the maturation process.

Authors’ response: Changed “balance” to “relationship”

lines 84, 85. The authors mean Brianna and Frontenac gris grapes, wines, or both?

Authors’ response: Added “fruit” to indicate that the grapes have the aroma attributes.

lines 111-114 : " Multivariate analysis was used to illustrate the effects of harvest time on wine odor active compounds. This is the first report of wine aromas made from Frontenac gris and Brianna grapes at different levels of fruit maturity. Information from this study can guide growers and winemakers in optimizing winemaking techniques and harvest decisions"  HAS ALREADY BEEN WRITTEN IN PREVIOUS PARAGRAPH

Authors’ response: we deleted the redundant text. 

lines 117-118: "berry maturation chemistry". Maybe the authors mean "berry maturation process"

Authors’ response: Deleted “chemistry” and added “..(i.e., change in pH and sugar content as °Brix)

lines 120-121: "Northern Grapes Project". Give a reference for that

Authors’ response: We changed line 121 to “… growing season and described as follows.”  Added website reference to Northern Grapes Project as reference number [19].  Citations were renumbered accordingly in the manuscript.

line 124: how many kgs? of grapes

Authors’ response: we added ‘(110-120 kg)’.

line 124: correct "distemmer" to "destemmer"

Authors’ response: Changed “distemmer” to “destemmer”

lines 124-125: why was the sugar content adjusted to different brix levels?

Authors’ response: We clarified this in the few additional sentences (to provide the same starting sugar content, resulting in the same ethanol production level across all treatments).

lines 124-125: If it is well understood, the must wasn't submitted to cold-settling after pressing and prior to fermentation. This might be a major reason for the "sulfuric, rotten egg" odors found in wines

Authors’ response: the must wasn't submitted to cold-settling. We added “The must wasn't submitted to cold-settling and might be a major reason for the ‘sulfury, rotten egg’ odors found in the research wines” in the Discussion.

line 128: Table Title. Correct to "Brianna and Frontenac Gris grapes characteristics at harvest"

Authors’ response: Corrected Table 2 title per reviewer’s request.

Also correct in the text when referenced.

Authors’ response: Reference to Table 2 in text was corrected per reviewer’s request.

Line 149: was agitation performed during incubation or prior to? explain better

Authors’ response: The agitation speed was for incubation and extraction, added text “during incubation and extraction”.

line 302: "throughout fermentation are" omit "are"

Authors’ response: Deleted “are” per reviewer’s request.

line 328: correct "are not able to" to "are not ALWAYS able to'"

Authors’ response: Edited text per reviewer’s request.

LINE 331: "To date, the only other published research on cold-hardy wine aromas pertains to red wines [41-44]." the authors need to take into consideration a master thesis

 "Descriptive analysis of Frontenac gris and Brianna wine grape and wine varieties "
A Thesis SUBMITTED TO THE FACULTY OF UNIVERSITY OF MINNESOTA
BY
Jenna M. Brady
December 2017

https://conservancy.umn.edu/bitstream/handle/11299/194662/Brady_umn_0130M_18804.pdf?sequence=1
Authors’ response: We added this reference.  Thank you.